# Lens Epithelial Explants Treated with Vitreous Humor Undergo Alterations in Chromatin Landscape with Concurrent Activation of Genes Associated with Fiber Cell Differentiation and Innate Immune Response

**DOI:** 10.3390/cells12030501

**Published:** 2023-02-03

**Authors:** Anil Upreti, Stephanie L. Padula, Jared A. Tangeman, Brad D. Wagner, Michael J. O’Connell, Tycho J. Jaquish, Raye K. Palko, Courtney J. Mantz, Deepti Anand, Frank J. Lovicu, Salil A. Lachke, Michael L. Robinson

**Affiliations:** 1Cell, Molecular and Structural Biology Program, Miami University, Oxford, OH 45056, USA; 2Department of Biology and Center for Visual Sciences, Miami University, Oxford, OH 45056, USA; 3Department of Statistics, Miami University, Oxford, OH 45056, USA; 4Department of Biological Sciences, University of Delaware, Newark, DE 19716, USA; 5Molecular and Cellular Biomedicine, School of Medical Sciences, and Save Sight Institute, The University of Sydney, Sydney, NSW 2006, Australia; 6Center for Bioinformatics and Computational Biology, University of Delaware, Newark, DE 19716, USA

**Keywords:** ATAC-seq, RNA-seq, lens, development, differentiation, explants, immune, motif, chromatin

## Abstract

Lens epithelial explants are comprised of lens epithelial cells cultured in vitro on their native basement membrane, the lens capsule. Biologists have used lens epithelial explants to study many different cellular processes including lens fiber cell differentiation. In these studies, fiber differentiation is typically measured by cellular elongation and the expression of a few proteins characteristically expressed by lens fiber cells in situ. Chromatin and RNA was collected from lens epithelial explants cultured in either un-supplemented media or media containing 50% bovine vitreous humor for one or five days. Chromatin for ATAC-sequencing and RNA for RNA-sequencing was prepared from explants to assess regions of accessible chromatin and to quantitatively measure gene expression, respectively. Vitreous humor increased chromatin accessibility in promoter regions of genes associated with fiber differentiation and, surprisingly, an immune response, and this was associated with increased transcript levels for these genes. In contrast, vitreous had little effect on the accessibility of the genes highly expressed in the lens epithelium despite dramatic reductions in their mRNA transcripts. An unbiased analysis of differentially accessible regions revealed an enrichment of cis-regulatory motifs for RUNX, SOX and TEAD transcription factors that may drive differential gene expression in response to vitreous.

## 1. Introduction

The exquisite and relatively simple design of the transparent ocular lens has made this unique tissue a source of experimental interest for well over a century. Surrounded by a collagen IV-rich capsule, the lens remains isolated from other cell types and is not innervated or vascularized. Within the confines of the lens capsule, the anterior hemisphere of the lens is lined by a subcapsular layer of cuboidal epithelial cells. Above the equator, a band of epithelial cells provides the proliferative force that fuels the life-long growth of the lens. At the lens equator, epithelial cells begin the morphological transition into elongated secondary fiber cells that make up the bulk of the lens structure. Although the central epithelial cells in the adult lens normally remain quiescent, the entire epithelium retains the capacity for proliferation and subsequent fiber cell differentiation. Lens epithelia undergo enormous physical and biochemical changes during their differentiation into fibers, including apical-basal elongation, elaborate lateral interdigitation, the accumulation of high concentrations of specific crystallin proteins and the loss of all membrane-bound organelles [1,2,3]. In its native state, maintenance of lens polarity depends on the influence of the ocular media; the aqueous humor on its anterior (epithelial) surface and the vitreous humor on its posterior (fiber mass) surface [4,5].

Many different biomedical disciplines have benefited from studies conducted on the lens. Spemann, in 1901, exploited the lens as the first model of embryonic induction when he showed in the frog, *Rana fusca*, that the early formation of the lens from the head surface ectoderm required the underlying optic vesicle [6]. The external accessibility and stereotypical, continuous pattern of epithelial proliferation and fiber cell formation has made the lens a particularly useful tissue for studying cellular differentiation. The ability of some salamanders to regenerate fully functional lenses from pigmented cells of the dorsal iris has contributed much to our understanding of cellular reprogramming [7,8]. A small upstream (~400 bp) region of the mouse αA-crystallin gene can drive lens-specific gene expression [9] and this made the lens a popular tissue for some of the earliest transgenic studies in mammals. The first demonstration of the use of Cre recombinase for tissue-specific gene deletion in mammals also utilized the lens [10]. Though naturally resistant to cancer, the lens can form tumors when both tumor suppressors Rb and p53 are inactivated [11,12]. This observation spurred numerous investigations of cell cycle regulation using the lens as an experimental platform [13]. From a medical perspective, cataract represents the most common lens disease and results from any condition that causes the loss of lens transparency. Despite highly successful surgical treatment for cataracts, the epithelial to mesenchymal transition (EMT) of lens epithelial cells that remain following cataract surgery frequently lead to posterior capsular opacification (PCO), the most common complication of this procedure [14,15]. The fibrotic EMT response of lens epithelial cells following cataract surgery has many similarities to the pathogenic pathways leading to numerous human diseases [16,17].

The lens epithelial explant system, initially established using embryonic chick epithelia, provided a way to observe fiber cell differentiation in vitro [18]. The creation of lens epithelial explants requires the removal of the lens fiber cells mass prior to pinning the lens capsule, with adherent epithelial cells, flat onto a culture dish. In contrast to primary dissociated lens epithelial cells or cultured lens epithelial cell lines, explanted lens epithelial cells retain their lateral attachments to neighboring cells as well as to their native basement membrane [19]. The ability to easily manipulate culture conditions made the lens epithelial explant system an extremely useful model to uncover the factors that promote lens fiber cell differentiation or those that lead to lens EMT and fibrosis [20,21,22,23,24,25,26]. The lens epithelial explant system was first applied to mammals by the McAvoy laboratory who showed that retina-conditioned media induced elongation and the accumulation of fiber-specific β- and γ-crystallins in explanted rat lens epithelium [27]. Later, this same group demonstrated that vitreous humor could effectively induce these changes as well [28]. Abundant evidence suggests that fibroblast growth factors (FGFs) are the most potent source of fiber differentiating activity of retina-conditioned media [29] and vitreous humor [30] in explanted lens epithelial cells; however, vitreous clearly contains other fiber cell-promoting activity beyond that of FGF alone [30,31,32].

The vitreous humor occupies the space between the lens and retina in the eye. While the vitreous humor is mostly water, it also contains abundant collagen fibrils, hyaluronic acid, and a host of other proteins and peptides [33]. Given the close proximity to both the lens and retina, it is not surprising that the vitreous humor would contain substances secreted by both tissues. In addition to FGFs, the vitreous humor contains other ligands for receptor tyrosine kinases including IGF1 [34], PDGF [35], EGF [36], HGF [37], VEGF [38,39] and EphrinA1 [33]. The vitreous also contains precursors of active TGFβ, CTGF and GDF [33] as well as BMP [40], all ligands of the TGFβ receptor superfamily. The levels of these signaling molecules within the vitreous varies with the physiological state of the eye and all have the potential to impact fiber cell differentiation, largely due to their ability to modulate ERK or SMAD activity [30,41,42,43,44].

Cell elongation and the appearance of β- and γ-crystallins in lens epithelial cells once defined the fiber cell differentiation process; however, modern transcriptomic technologies have now made it possible to distinguish lens epithelial cells from lens fiber cells in a much more thorough way. RNA-seq studies have comprehensively cataloged genes that are differentially expressed between lens epithelial and fiber cells in the developing mouse [45,46] and chick [47]. More recently, a genome-wide analysis of chromatin accessibility by ATAC-seq revealed differential regions of opened and closed chromatin that define these two cell types during mid-gestation and neonatal mouse lens development [48]. Similarly, RNA-seq and ATAC-seq studies have compared lens epithelial cells and fiber cells in the developing chick embryo [49]. Analysis of single cell RNA-seq in zebrafish [50] and single nuclei RNA-seq in humans [51] suggests a continuum of lens cell states, rather than discrete gene expression differences along a differentiation pathway from anterior lens epithelium to mature fibers.

Given the ability to extensively analyze both gene expression and chromatin accessibility, rather than select defined differentiation markers, we sought to analyze lens fiber cell differentiation in the mouse lens epithelial explant system induced by vitreous humor. To do so, we conducted both RNA-seq and ATAC-seq analyses of cultured lens epithelia cultured with or without vitreous humor, an established inducer of fiber cell differentiation in lens epithelial explants [28,30,32]. This global analysis of gene expression modulation—going beyond a select few markers—stands to comprehensively inform on the genome-wide changes that are associated with in vitro fiber cell differentiation.

We have leveraged ATAC and RNA sequencing of cultured lens epithelia to characterize the genome-wide regulatory landscape of this historically valuable model of fiber cell differentiation. As expected, vitreous treatment induced a sharp repression of epithelial-identity genes, such as *Foxe3*, *Gja1*, and *Cdh1*. Conversely, the expression of fiber cell-identity genes such as *Cryba4*, *Hsf4*, and *Crybb1* were upregulated throughout time in culture. However, many fiber cell genes were only modestly increased in vitreous-treated explants relative to media-treated controls, suggesting that vitreous may exert its early pro-differentiation effects primarily through the suppression of epithelial identity, rather than the activation of fiber identity. Our findings expound promoter regions which undergo substantial expansions in accessibility in response to vitreous, many of which are known fiber cell-related genes, such as *Tmod1*, *Gja8*, and *Lgsn*. Paradoxically, we identify a broad set of gene promoters associated with epithelial identity that remain in a highly accessible state following vitreous treatment, despite dramatic reductions in their expression. Examples of such genes include *Dll1*, *Fzd1*, and *Kdr*. This suggests that there could be lingering vestiges of epithelial identity found within the chromatin organization of lens explants, even after 5 days of vitreous exposure. This should be closely considered by others in this field as they make use of this model system. Vitreous treatment induced broad activation of inflammatory and EMT gene sets within 5 days, as well as accessibility footprints indicative of AP1 binding activity, pointing toward widespread activation of stress-responsive factors. Finally, we identify broad signatures of RUNX1 binding activity in explants treated with ‘differentiation’ media and verify the enhanced expression of *RUNX1* in human lens fiber cells, suggesting a novel role for this factor in the establishment and maintenance of fiber cell identity. Collectively, our results provide an unprecedented portrait of the dynamic chromatin architecture within vitreous-treated lens epithelia and serve as a valuable resource to researchers seeking to wield this insightful model.

## 2. Materials and Methods

### 2.1. Animals

All procedures were approved by the Miami University Institutional Animal Care and Use Committee and complied with the ARVO Statement for the Use of Animals in Research, consistent with those published by the Institute for Laboratory Animal Research (Guide for the Care and Use of Laboratory Animals). *FVB/N* mice were euthanized at postnatal day 8 (P8) by CO_2_ asphyxiation followed by cervical dislocation.

### 2.2. Lens Epithelial Explant Generation and Culture

Mouse lens epithelial explants were prepared from postnatal day 8 (P8) mice as previously described [19,32]. Lenses were removed and placed into a 35 mm culture dish with pre-calibrated (37 °C and 5% CO_2_) M-199 media supplemented with 1% antibiotic-antimycotic solution (ThermoFisher Scientific cat. 15240062) and 0.1% bovine serum albumin (Sigma-Aldrich cat. A7906) henceforth referred to as “culture medium (CM)”. The lens capsule was torn at the posterior pole prior to removing the fiber cell mass. The remaining lens capsule with adherent epithelium was pinned flat to the base of the culture dish. Fresh bovine vitreous was collected as previously described [28] (courtesy of Kaiser Meat Market, Cedar Grove, IN, USA) prior to storage at −20 °C. For all treatment groups other than the Immediate Collection group, explants were cultured in culture media for 24 h after generation. Following this, some treatment groups were cultured in vitreous, diluted 50% (*v*/*v*) with culture media, henceforth referred to as “differentiation medium (DM)”, for 1 day or 5 days.

### 2.3. RNA-Seq Library Preparation and Sequencing

RNA-seq analyses were carried out for each condition with three biological repilcates. Each replicate consisted of 12 explants. Explants were collected in 1.5 mL Eppendorf tubes. Total RNA was extracted using the RNEasy Mini Kit (Qiagen cat. 74104) following the manufacturer’s instructions. The RNA Integrity Number (RIN) was determined using an Agilent 2100 Bioanalyzer and samples with RIN > 7 were used for sequencing. Samples ranged from 400 ng to 1 µg of total RNA. RNA libraries were constructed by DNA Link (Los Angeles, CA, USA) using a TruSeq (Illumina) library prep kit. The library was sequenced on the Illumina NovaSeq 6000 sequencer platform. Sequences consisted of 100 base paired-end reads to a minimum depth of 26 million reads per sample.

### 2.4. RNA Seq Data Analysis

Raw reads were quality-analyzed using FastQC (Babraham Bioinformatics- FastQC A Quality Control tool for High Throughput Sequence Data) [52,53] and MultiQC [54]. Low-quality bases and adapters were trimmed using Cutadapt 3.4 [55] and Trim Galore 0.6.5-1 with the parameters -q 20 --phred33 --length 20 [56]. Mouse genome GRCm39 version: M27 was indexed using Hisat2 (2.1.0-4) [57], incorporating splice junctions from Gencode GTF gencode.vM27.annotation.gtf file [58]. Gene counts were generated using Stringtie2.1.5 [59] and gencode GTF annotation gencode.vM27.annotation.gtf. Differential expression testing was performed with DESeq2 [60]. Differentially expressed genes (DEGs) are defined throughout by an adj. *p*-value ≤ 0.05, and log_2_ fold change (LFC) ≥ 1.5 criteria were applied. In these cases, the *p*-value was adjusted using the Benjamin-Hochberg correction individually for each pairwise comparison made.

Clustering of DEGs was performed with K-means clustering using normalized RNA-seq counts [61]. Specifically, the kmeans function (part of the “stats” package in R) was used with the Hartigan-Wong algorithm [62,63]. Pathway enrichment analysis was performed with gProfiler (R package) [64]. All individual clusters obtained from K-means clustering were used for GO term analysis in groups to reduce biases. This analysis resulted in *p*-values associated with each cluster for each identified GO term, facilitating consideration of biological functions associated with each cluster. The obtained *p*-values were −log_10_ normalized prior to visualization as a bar chart. All figures were generated in the R environment.

Gene set enrichment analysis (GSEA) was performed on the normalized count matrix obtained from DESeq2, and murine genes were converted to human orthologs [65,66]. GSEA was performed using 1000 permutations and gene set permutations with gene set size filters; min = 15 and max = 500. Hallmark gene set was used for analysis. The newborn *FVB/N* mouse lens epithelium and fiber data were downloaded from the GEO database with the SRA accessor: SRP040480 [45].

### 2.5. RT-qPCR Validation of RNA-Seq Data

Select genes identified by RNA-Seq were validated using RT-qPCR. RNA was extracted as described above, and samples were reverse transcribed into cDNA using oligo (dT) primers and random primers using ImProm-II^TM^ Reverse Transcription System (A3800), according to manufacturer’s instructions. The RT-qPCR assays were performed in triplicate biological replicates and triplicate technical replicates using GoTaq qPCR Master Mix (A6001) and the Bio-Rad CFX Connect. Pre-designed primers from primerbank (https://pga.mgh.harvard.edu/primerbank/, accessed on 18 January 2023) were ordered and obtained from IDT technologies (Appendix A). The comparative ΔΔCt method was used to determine relative gene expression levels compared to housekeeping genes (*Eef1a1*) and relative mRNA was normalized to immediately collected explants (IMD). Prior to statistical analysis, fold change values were log-transformed in order to promote conformity to modeling assumptions. One-way ANOVA was performed in R using the stats package [63], and Levene’s test was applied using the car package [67] to assess homogeneity of variances. Following ANOVA, means and standard error were computed using the R package emmeans [68]. For samples with 3 groups, Tukey’s Honest Significant Difference test was used post-hoc to calculate adjusted *p* values [63]. One gene, *Runx3*, was expressed below the threshold of detection of the assay in a single condition. For *Runx3*, a two-sample equal variances *t*-test [63] was used to calculate *p* value.

### 2.6. ATAC-Seq Library Preparation and Sequencing

ATAC-seq library preparation was performed using the ATAC-Seq Kit (Active Motif cat.53150) per the manufacturer’s instructions. Briefly, four lens epithelial explants per sample (biological replicate) were collected in ice-cold PBS. Two biological replicates were conducted for each ATAC-seq condition. The samples were centrifuged at 8000× *g* for 5 min at 4 °C. 100 µL of ATAC lysis buffer was added to the samples after aspirating the PBS. The explant samples were pipetted vigorously until visibly broken. The total number and quality (round shape without blebbing) of isolated nuclei was accessed by trypan blue staining that ranged between 80,000–120,000 per sample. Following nuclei validation, the samples were centrifuged, tagmented and PCR amplified. The amplified DNA was purified and indexed (Dual indexing following manufacturer protocol) to create sequence-ready libraries. The libraries were quantified using a Qubit 4 Fluorometer with the dsDNA Quantitation, high sensitivity kit (ThermoFisher Scientific, cat. Q32851). Final libraries were quality validated using the Bioanalyzer HS DNA Kit (Agilent, cat. 5067-4626). Validated libraries passing in-house quality control were sent to Novogene (Sacramento, CA, USA) for sequencing with a HiSeq platform using 150 base paired-end reads to a depth of greater than 30 million reads.

### 2.7. ATAC-Seq Data Analysis

Raw reads were trimmed using Cutadapt 3.4 and Trim Galore 0.6.5-1 with the trimming parameters -q 20 --phred33 --length 20. Alignment of raw reads to the *Mus musculus* genome GRCm39 was performed with Bowtie2 [69] using the parameters -X 2000 --very-sensitive-local --no-mixed --no-discordant --no-dovetail. Peaks were called for samples individually with MACS2 [70] using parameters -f BAMPE -g mm -q 0.05. Peaks aligning to the mitochondrial chromosome were discarded. Differential accessibility testing of peak regions and peak summits was performed with DESeq2. Differentially accessible regions (DARs) identifed by DESeq2 were those with a Log_2_ fold change of ≥1.5 and adjusted *p*-value (*p*-adjust) of ≤0.05. ATAC coverage tracks were visualized using the Integrative Genomics Viewer (IGV) [71]. Prior to visualization, BAM files were converted to BIGWIG format and normalized using the bamCoverage --binSize 1 and --scaleFactor parameter using the inverse of DESeq2 size factors. Accessibility heatmaps were generated using deepTools computeMatrix reference-point and plotHeatmap functions [72].

### 2.8. Genomic Annotation and Motif Prediction

Peak annotations to the nearest TSS were performed using ChIPseeker [73]. Binned motif analysis was executed using the monaLisa R package [74] and the JASPAR 2022 vertebrate motif database [75]. For monaLisa analysis, all differentially accessible peak summits (|LFC| ≥ 1.5 and adj. *p*-value ≤ 0.05) between conditions were identified. This was further partitioned into differentially accessible regions (DARs) regulated by DM and CM from 1 or 5 days. Finally, the subset regions (promoter vs. remaining accessible regions) were subjected to motif prediction using the monaLisa R package. Occurrences of motifs of interest were annotated back to the moue genome using the monaLisa function findMotifhits. Furthermore, the difference in the distribution of the genomic annotation of promoters and remaining regions were analyzed using ChiPseeker. The identified motif annotated regions were manually verified using IGV. Venn diagrams were made using Venny [76].

### 2.9. Single Cell Data Analysis

The single cell data on human ocular anterior segment [51] was accessed and visualized in the Broad Institute’s Single cell portal (https://singlecell.broadinstitute.org/single_cell/study/SCP1841/, accessed on 19 September 2022). The dot blot and TSNE plots were downloaded from the same portal.

### 2.10. Relationship between ATAC-Seq and RNA Seq

To compute the overall correlation between the ATAC-seq and RNA-seq data, the normalized counts were computed for both data sets using DEseq2 in R. Then the ATAC-seq peaks were matched to genes based on the gene with the nearest promoter. For genes with multiple nearby ATAC-seq peaks, the ATAC-seq counts were averaged. Then the Pearson correlation coefficient was computed for average normalized counts across all samples for all genes. 

To compare the results from the clustering analysis of the differential gene expression and differentially accessible regions, the ATAC-seq peaks were once again matched to the genes with the nearest promoters. Then, a cross-table was generated to indicate the number of genes in each pair of RNA-seq and ATAC-seq clusters. A Chi-squared test of independence was run to determine whether the cluster sets were independent. A permutation-based analysis of this table indicated which cluster pairs were associated with each other. Using fixed cluster sizes, the RNAseq and ATACseq clusters were randomly permuted 100,000 times. For each permutation, a new cross-table was generated, and the permuted cell counts were compared to the observed cell counts. Clusters were considered to be associated if less than 5 percent of the permuted counts for that cell were greater than or equal to the observed count.

To determine a potential relationship between basal accessibility and transcriptional output, all annotated mouse transcripts were ordered by average expression across biological replicates within each condition. Ordered loci were used as input for the deepTools2 command line suite [72]. The functions computeMatrix and plotHeatmap were used to plot heatmaps of normalized ATAC signal for transcripts within each quintile of expression bin.

## 3. Results

### 3.1. Chromatin Accessibility and Gene Expression Changes in Lens Epithelial Explants Resulting from Explanting, the Addition of Vitreous, and Time in Culture

To analyze how the process of in vitro culture with and without the addition of bovine vitreous changes chromatin accessibility and gene expression in lens epithelial explants, we harvested lens epithelia from postnatal day 8 (P8) mice and explanted the epithelium in a culture dish supplemented with either culture media (CM) or differentiation media (DM). Following lens explant culture, samples were harvested at five specific time points for the collection of chromatin and RNA. These were (A) immediately after collection of the epithelium (IMD), (B) after 24 h in CM (Day 0), (C) after an additional day in CM (D1_CM) or DM (D1_DM), (D) after an additional (relative to Day 0) 5 days in CM (D5_CM) or DM (D5_ DM) (Figure 1A). Our experimental design consists of many variables that can affect both chromatin organization and gene expression. These include: (i) the trauma induced by the physical removal of the epithelium and the establishment of in vitro culture, best represented by the D0 treatment; (ii) the time in culture, represented by the D1 and D5 samples and (iii) differential effects of unsupplemented culture medium and the addition of vitreous, represented by the differences between the CM and DM treated samples. We reasoned that day 0 (D0) would best reflect immediate changes resulting from in vitro culture, while D1 would reflect immediate differences resulting from the addition of vitreous in the DM condition. In previous experiments in neonatal rat lens epithelial explants, immunologically detectable β-crystallin expression was evident after five days of 50% vitreous treatment [30].

We performed three-dimensional principal component analysis (3D-PCA) on both the ATAC-seq (Figure 1B) and the RNA-seq (Figure 1C) datasets obtained. The first three components PC1, PC2 and PC3 were plotted as X, Y and Z axes to observe the major components shaping each dataset. For both the ATAC-seq and the RNA-seq PCA plots, the X axis represents the primary dimension of variation (PC1) and clearly separates the DM-treated samples from the other samples. With respect to the ATAC-seq data, with two replicates per treatment, and the RNA-seq data, with three replicates per treatment, the IMD replicates occupy nearly the same space on each of the plots. Moreover, the replicates within each treatment are consistently clustered together on both the ATAC-seq and RNA-seq PCA plots, underscoring a high concordance between our biological replicates. The overall similarity of the biological replicates and the effect of treatments can be alternatively visualized by a distance matrix heatmap for the ATAC-seq data (Appendix A) and the RNA-seq data (Appendix A).

### 3.2. Clustering of ATAC-Seq and RNA-Seq Data to Evaluate Effects of In Vitro Culture, Time in Culture, and the Addition of Vitreous Humor on Chromatin Organization and Gene Expression

We sought to generate an overview of meaningful changes in gene expression in response to our tested treatments. To this end, we employed a series of pairwise comparisons that stand to capture changes in expression that are relevant to our questions of interest. To examine the effect of explanting only, we analyzed IMD vs. D0. In addition, we compared both the IMD and D0 treatments to the CM and DM treated groups at both time points, which revealed genes responsive to the effects of culture and vitreous relative to our baseline conditions. Finally, we tested all possible pairwise combinations of our CM and DM treated explants at both times in culture, which revealed genes that were undergoing changes in response to vitreous, time in culture, or both. From these comparisons, we then generated a non-redundant list of differentially expressed genes and performed K-means clustering to organize them into clusters representing archetypal expression patterns within our dataset. These comparisons identified differentially accessible regions (DARs) and differentially expressed genes (DEGs) from the ATAC-seq and RNA-seq datasets, respectively.

The K-means clustering of DARs from the pairwise comparisons of the treatments revealed six clusters of genomic coordinates encompassing a total of 129,635 distinct genomic fragments that changed in specific ways (Figure 2A and Appendix A). After defining these clusters, we used Chipseeker to assign genes to each DAR based on the nearest promoter region(s). This information made it possible to search for gene ontology (GO) terms associated with each cluster to assess the biological significance of the changes in chromatin accessibility (Appendix A). This analysis made it easier to identify how the genomic landscape changed in lens explants with both time in culture and with differential media (representative GO terms are shown in Figure 2B). Cluster A (22,965 sites) represents regions that become more accessible immediately after explanting (D0) and after long term culture in unsupplemented media (D5_CM) but generally becomes less accessible with DM. Cluster A was enriched with GO terms for inflammatory process, and MAPK cascade. Cluster B (19,103 sites) exhibited the opposite accessibility profile of cluster A and was enriched for actin filament-based process, and actin cytoskeletal organization. Cluster C (20,723 sites) exhibited the highest accessibility in the D5_DM.

Samples were enriched for GO terms such as regulation of cell migration, and transcription repressor complex. Cluster D (30,694 sites) exhibited the highest accessibility in the IMD samples and lowest accessibility in the D5_DM samples treatments and was enriched for GO terms including cell morphogenesis involved in differentiation, and the calcium signaling pathway. Cluster E (21,240 sites) represented regions that are inaccessible in the IMD samples with increased accessibility in both D0 and in DM treated samples. Cluster E was enriched for GO terms like positive regulation of cell differentiation, and cell-substrate adhesion. Cluster F (14,910 sites) exhibited the highest accessibility in D1_CM samples and this cluster was enriched for terms including extracellular matrix, and cell morphogenesis involved in differentiation. These observations suggest that each treatment results in a dynamic shift in epigenetic landscape, with the potential to impart functional significance by impacting nearby transcription.

The effects of explanting and differential media appeared to have more impact on chromatin accessibility than did time in culture. DAR clusters B, C and E generally gain chromatin accessibility in response to vitreous addition, and of these, cluster E represents the most dramatically different from the accessibility profile of the native lens epithelium represented by the IMD sample. In contrast, DAR cluster D exhibited largely accessible chromatin in the IMD samples that become distinctly less accessible in the DM media, particularly at day 5. DAR clusters A and F exhibited the highest accessibility in the D5_CM and D1_CM conditions, respectively. DAR clusters A and E represented regions that gained accessibility within 24 h of explanting while clusters B, D and F lost accessibility at this time point. All clusters exhibited nearly opposite accessibility profiles in the CM versus DM conditions. DAR clusters B, C and E exhibited higher accessibility in the DM condition while clusters A, D and F were less accessible in the DM than the CM condition. In no case did a cluster exhibit a similar trend in accessibility with time in the DM and CM conditions.

As we did for the ATAC-seq data, we used K-means clustering to independently analyze the RNA-seq data from each of the fifteen pairwise comparisons described above and performed GO analysis to the genes assigned to each cluster. The K-means clustering of DEGs also revealed six distinct clusters from a total of 9044 DEGs (Figure 2C, Appendix A). Selected GO terms for each cluster are shown on a bar plot (Figure 2D) with the full list of GO terms associated with each cluster shown in Appendix A. Cluster 1 (1098 genes) represented genes with high expression in the IMD and D5_DM samples and low expression in the D0 samples. Cluster 1 is enriched with genes associated with lens development (*Bfsp1*, *Mip*, *Lim2*, *Hsf4*), actin cytoskeleton organization (*Myo5c*, *Myo7a*, *Rhoh*, *Rac2*) and regulation of the immune system (*Tnf*, *Ccr1*, *Csf1r*, *Tlr7*). Cluster 2 (1291 genes) represented genes with low expression in the D5_CM samples, high expression in the D1_DM samples, and moderate expression in the IMD and D5_DM samples. Cluster 2 was enriched for genes associated with chromosome segregation (*Cenpn, Spag5, Aurkb, Bub1b*), the cell cycle (*Cdc6*, *Cdk1*, *Ccne1*, *Pcna*) and mitosis (*Birc5*, *Ube2c*, *Cdc14a*). Cluster 3 (1731 genes) represented genes with high expression in the IMD samples and was associated with several terms suggestive of chromatin reorganization, including those associated with the polycomb repressive complex 2 (*H2ac6*, *H4C9*, *H2bc11*, *H2bc12*) and histone acetylation/deacetylation (*H2bc3*, *H2bc15*, *H3c14*, *H4f16*) and gated channel activity (*Gabra2*, *Kcnq1*, *Scna4*, *Clcn2*). Cluster 4 (1477 genes) represented genes with high expression in the D1_DM and D5_DM samples and was highly enriched for cell migration and epithelial to mesenchymal transition (EMT) genes (*Mmp7*, *Fn1*, *Col1a1*, *Snal1*). Cluster 5 (1062 genes) represented genes with low expression in the IMD samples and moderate expression in all the explanted samples including several genes associated with phagocytosis (*Rab7b*, *Slc11a1*, *Tap1*, *Clec4e*). Cluster 6 (2385 genes) represented genes with high expression in the IMD samples and samples treated with CM but low in DM treated samples. This cluster contained several genes associated with neuronal development (*Mapt*, *Map2*, *L1cam*, *Trpm1*).

Similarly, clustering of DEGs revealed that the addition of vitreous to cultured lens epithelia upregulated many genes associated with fiber cell differentiation and immune system processes and downregulated many genes highly expressed in the lens epithelium. DEG clusters 1, 2 and 4 exhibited specific upregulation upon the addition of vitreous (D1_DM and D5_DM). Of these, the DEGs in cluster 2 exhibited specific upregulation at D1_DM and were enriched with genes associated with proliferation and the upregulation of MAPK signaling (*Cdk1*, *Myc*, *Spry4*, *Etv1*, *Etv4*, *Etv5*), suggesting an immediate response to the addition of growth factors present in the vitreous humor. In contrast, DEG cluster 1 exhibited specific upregulation in the D5_DM condition. This cluster contained most of the crystallin genes associated with fiber cell differentiation (*Crybb1*, *Crybb3*, *Cryba4*, *Crygb*, *Crygc*, *Crygd*, *Cryge*, *Crygs*) as well as other genes highly expressed in fiber cells such as *Nav3*, *Mip*, *Lim2*, *Hsf4* and *Dnase2b*. Cluster 4 also contained genes highly expressed in fiber cells (*Pla2g7*, *Slc2a3*, *Jag1* and *Lgsn*) but this cluster also contained a large number of genes associated with extracellular matrix, fibrosis and immune response (for example: *Tgbi*, *Fn1*, *Mmp7*, *Mmp9*, *Col5a3*, *Ccl6*, *Ereg*, *Axl*, *Csf3*, *C3*, *Krt16*, *Cd44*). Clusters 3 and 6 exhibited high expression in the IMD, lens epithelium samples, but lower expression in all other conditions. Cluster 6, in particular, exhibited significant downregulation upon the addition of vitreous (D1_DM and D5_DM). In agreement with this, both clusters contained genes highly expressed in the lens epithelium. These included notch ligands (*Dll4* and *Dll3*) and *Rgs6* in cluster 3 and epithelial genes including *Foxe3*, *Kdr*, *Sulf1*, *Pdgfra*, *Cdh1*, *Gja1*, *Npnt* and *Podn* in cluster 6.

An overall correlation between the number of differentially expressed genes and differentially accessible regions was analyzed for each of the cluster for RNA-seq and ATAC-seq (Table 1). The strongest association was observed for ATAC cluster CD and RNA-seq cluster C6.

Furthermore, concordance of the ATAC seq signal with gene expression was analyzed for the D1 and D5 data (Table 2) to further dissect the relation between the datasets (Table 2). The highest concordance was observed between the accessible regions and the corresponding gene expression for D5_DM samples.

We analyzed the relationship of chromatin accessibility at transcription start sites (TSS) with gene expression throughout the genome in all experimental conditions. We uniformly observed a high agreement between basal gene expression and TSS accessibility (Figure 3). The normalized count matrix for the entirety of the ATAC-seq accessible regions and RNA-seq gene expression datasets were used to calculate the correlation between ATAC-seq and RNA-seq. This analysis led to an r value of 0.404 (Appendix A).

We next sought to determine whether genes that exhibit expression patterns that are linked to promoter region accessibility could be associated with specific biological processes that explain the response to vitreous. To do this, we categorized DEGs observed after 5 days of DM treatment into categories based on the degree of correlation between gene expression and promoter region accessibility (Appendix A). Interestingly, genes which exhibited a strong positive correlation (*n* = 115) were associated with apelin signaling. A large number of genes exhibited a moderate positive correlation between expression and promoter accessibility (*n* = 2062), and were strongly enriched for genes associated with mitosis, epithelial proliferation, mesenchymal cell differentiation, and fibroblast proliferation. In contrast, the genes without any detectable correlation were related to diverse functions, including organelle structure, endosomes, microtubule organization, rRNA metabolism, actin filaments, and supramolecular fiber organization. This set of genes could collectively represent a broad set of processes regulated by vitreous for which promoter accessibility is not a primary driver of gene expression changes. We observed a relatively smaller number of genes which exhibited moderate negative (*n* = 1152) or strong negative (*n* = 27) correlation with gene expression, which may reflect the paradoxical nature of an opposing relationship between promoter accessibility and expression. Genes with moderate negative correlation tended to overlap many of the processes observed for genes with no correlation, suggesting a functional similarity between genes of these categories.

We selected a list of 90 genes in different categories to look at more closely in order to discern more about the biology and developmental state of our explants upon culture and vitreous addition. These genes were based on those differentially expressed between lens epithelial cells and lens fiber cells [33] or on genes associated with particular GO terms. These genes were divided into five classes: (a) twenty-four immune response-related (Immune) genes, (b) twenty-four lens fiber cell-related (Fib) genes, (c) twenty lens epithelial cell-related (Epi) genes, (d) seven mitogen activated protein kinase-related (MAPK) genes and (e) fifteen extracellular matrix-related (ECM) genes. Both ATAC-seq and RNA-seq clusters associated with each of these genes were identified to see if there was any obvious correlation between the two cluster sets (Appendix A). With the exception of RNA cluster 6, the majority of the selected genes in all RNA clusters were in ATAC cluster A. For cluster 6, 27.8% of the genes were in ATAC cluster A and 61.1% of the genes were in ATAC cluster B. Although some genes in every RNA-seq cluster were assigned to ATAC cluster A, the largest proportion of the ATAC cluster A genes (47.1%) were in RNA cluster 4. RNA clusters 1 and 6 made up 35.3% and 64.7% of the genes in ATAC cluster B, respectively. Of the genes in ATAC cluster C, 78.6% came from RNA cluster 4. Few of the 90 selected genes were found in ATAC clusters D (1), E (2) and F (4) and none of these genes were found in RNA clusters 1 or 3.

### 3.3. Assessment of Genome-Wide Dynamic Changes in Chromatin Accessibility with the Addition of Vitreous Humor

To discern where changes in the chromatin landscape occur relative to genes, the accessible regions identified by ATACseq were annotated using CHIPseeker. CHIPseeker categorizes the accessible regions as associated with promoters, introns, exons, untranslated regions (UTRs), or intergenic. These regions are further broken down as proximal promoters (within 1 kb of the TSS), intermediate promoters (within 1–2 kb of the TSS) and distal promoters (within 2–3 kb of the TSS), first exon, other exon, first intron, other intron, 5′ untranslated region (5′ UTR), 3′ untranslated region (3′ UTR), downstream intergenic (within 300 kb of the 3′UTR), and distal intergenic (>300 kb of the 3′UTR) regions. We observed that the proportion of accessible chromatin regions corresponding to promoters decreased with time in DM-treated samples, while the proportion of distal intergenic accessible regions increased with time, relative to the IMD samples (Figure 4). In contrast, the CM-treated samples exhibited the opposite accessibility profile, with increased proximal promoter accessibility and decreased distal intergenic region accessibility with time relative to the IMD samples (Figure 4). We observe the same phenomenon if we focus only on the DARs between the CM- and DM- treated samples for each day (Appendix A). Again, the number of accessible promoters for CM media increased from 615 on D1 to 5973 on D5 (9.7 fold) whereas those in the DM explants increased from 1260 on D1 to 4054 on D5 (3.2 fold). However, the non-promoter accessible regions for CM-treated explants increased from 2442 on D1 to 16,954 on D5 (6.9 fold) and in the DM-treated explants the number of accessible non-promoter regions increased from 6950 on D1 to 20,493 on D5 (2.9 fold). Overall, DM treatment of lens epithelia leads to a less substantial increase in both promoter and non-promoter accessibility with time relative to the lens epithelial explants cultured in CM.

### 3.4. Specific Changes in Chromatin Accessibility and Gene Expression in Vitreous-Treated Lens Epithelial Explants with Respect to Fiber Cell Differentiation

Given the extensive use of lens epithelial explants to model lens fiber cell differentiation, we investigated how the chromatin landscape and transcription of genes associated with lens fiber cells changed in the different treatment groups. Analyzing the genomic regions associated with genes highly expressed in lens fiber cells, we observed genomic accessibility patterns near the promoter regions (Figure 5A and Appendix A). The promoters of some genes (for example: *Gja3, Bnip3, Hsf4, Grygn, Maf, Mip* and *Arid3b*) exhibited a constitutive accessibility pattern (no evidence of changes in accessibility in the IMD samples and in all treatments). A second pattern (exemplified by: *Gja8*, *Carhsp1*, *Grygs*, *Slc2a3*, *Elf3*, *Jga1*, *Dach2*, *Sox1*, *Crybb3*, *Cryba4/Crybb1*, *Lctl* and *Lim2*) exhibited a relative increase of accessibility in the IMD and DM treated conditions, but a loss of accessibility in the CM treatments. Several of the promoters in this group also exhibited a transient accessibility increase in the D1 CM that was lost by D5 (Appendix A). A third set of gene promoters (e.g.,: *Caprin2*, *Crygd*, *Fabp5*, *Cd24a* and *Mitf*) were accessible in the IMD samples, but exhibited increased accessibility with DM treatment. A fourth group of genes (e.g.,: *Tmod1* and *Bache*) had promoters that were relatively inaccessible in the IMD samples with increased accessibility in D0 and/or CM but exhibited peak accessibility in the DM treatments. Finally, a fifth group of promoters (e.g.,: *Lgsn*, *Dnase2b* and *Nav3*) did not show discernable/relative enrichment of ATAC signal in any treatments except the DM samples.

We observed interesting patterns in gene expression with respect to genes associated with fiber cell differentiation (Figure 5B). One group of genes that included *Tmod1*, *Pa2g7*, and *Slc2a3* exhibited very low expression in the P8 lens epithelium (IMD) and only reached high expression levels after 5 days of vitreous exposure. Therefore, high expression of these genes requires more than short term exposure to factors present in the vitreous humor. A second group of genes including *Elf3*, *Jag1*, and *Lgsn*, exhibited low expression in the IMD condition and high expression when exposed to vitreous starting at D1. This was the only group of fiber-cell genes that failed to exhibit a dramatic relative decline in expression after one day of vitreous exposure (D1_DM). The third gene expression pattern (characterized by *Gja8*, *Aldh1a1*, and *Eln*) exhibited high expression in IMD and D5_CM conditions and very low expression after one day of vitreous exposure that only increased slightly at 5 days. High expression of these genes appeared independent of vitreous during the culture period. Yet another class of fiber-cell genes, including *Crybb3*, *Cryba4*, *Crygs*, *Mip*, and *Hsf4*, underwent an unexpected, transient decrease in expression at D1 for both CM- and DM-treated samples. However, expression of these genes was increased to similar levels in both CM- and DM-treated samples, revealing that expression of these genes is enhanced independently of vitreous treatment within 5 days of culture.

The observed initial decline in fiber cell gene expression upon the addition of vitreous contrasts with the increased chromatin accessibility seen for most of these genes in the D1_DM samples. This observation mirrors the initial drop in crystallin transcript expression following the addition of FGF2 in rat lens epithelial explants [77]. Unsurprisingly, genes associated with MAPK signaling (*Etv1*, *Etv4*, *Etv5*, *Spry1*, and *Spry4*), upstream of fiber cell differentiation, exhibit an increase in expression following the addition of vitreous that is significantly higher in D5_DM than in D1_DM (Appendix A). This would be consistent with a requirement of sustained MAPK for the expression of several genes associated with fiber cell differentiation [78].

Fiber cell differentiation, while often characterized by the increased expression of genes and proteins preferentially found in lens fiber cells, also involves the loss of transcripts and proteins preferentially found in lens epithelial cells. We examined the chromatin accessibility of promoter regions of genes preferentially expressed in the epithelium to compare the accessibility in the native lens epithelium (IMD) with that in the DM-treated explants (Appendix A). For the majority of the genes we examined (14 of 22, including *Pax6, Pdgfra*, *Cdh1*, *Gja1*, *Fzd1*, *Emilin3*, *Dll1* and *Cdk1*) the accessibility near the transcription start site remained the same or declined slightly in the DM relative to the IMD samples. In a few cases (*Kdr, Mme*, *Tulp3*, *Npnt*, and *Igsf3*) the accessibility increased in DM, and in other cases (*Foxe3*, *Islr* and *Podn*) the accessibility significantly declined in DM relative to the IMD samples.

To explore how the lens transcriptome changes with our treatments with respect to genes that typify the lens epithelium, we selected 20 genes that are preferentially expressed in lens epithelial cells over lens fiber cells in newborn lenses [45]. The expression of most of these genes declined upon treatment with DM (Figure 5C). The typical pattern (exhibited by *Foxe3*, *Cdh1* and *Fzd1*) is moderate to high expression in the IMD samples, followed by a drop upon explanting (D0) with some recovery in CM and drastically reduced expression in DM. Some exceptions include the proliferation-related genes (*Cdk1* and *Mki67*) where long term culture in CM or DM resulted in expression declines, but the initial addition of vitreous (D1-DM) results in a dramatically increased expression. Likewise, some genes (for example, *Notch4* and *Flt1*) exhibited a reduced expression in D0 and CM but an increased expression in DM.

We also performed a statistical analysis of the top 30 most significant differentially expressed protein-coding genes between epithelial and fiber cells ranked by adjusted *p* value from a previous RNA-seq study on newborn mouse lens [45] by comparing the expression of these genes between the IMD and D5_DM condition. In the D5_DM samples, 20 of the top 30 “lens epithelial genes” were downregulated, while only 2 of these genes were upregulated. In contrast, only 5 of the 30 “fiber cell genes” were differentially regulated between the IMD and D5_DM condition but all of these were upregulated in DM. A Fisher’s exact test indicated a significant association between the gene sets (*p* = 0.0003).

### 3.5. Increases in Both Promoter Accessibility and Gene Expression for Epithelial to Mesenchymal Transition and Inflammatory Response Genes Induced by Vitreous Humor

We observed a very consistent pattern of increased accessibility in the promoters of genes associated with immune activation, inflammation, and fibrosis with the addition of DM (Appendix A). In many cases, these genes were inaccessible (for example, *Krt16*), or barely accessible (for example, *Tnfrs11b*) in the IMD condition (Figure 6A). Likewise, when examining a representative list of immune-response-related genes (Figure 6B) or genes associated with ECM organization (Appendix A), we typically found very low expression in the IMD samples with significant upregulation in DM media. We used gene set enrichment analysis (GSEA) to conduct a comprehensive pairwise comparison of the IMD versus D5_DM gene expression. This analysis utilized hallmark gene sets to identify differential sample enrichment for well-defined biological states. Among the most enriched states identified in the D5_DM sample were epithelial to mesenchymal transition and inflammatory response (Figure 6C). These findings provide compelling evidence that the addition of vitreous humor to lens epithelial explants leads to a pronounced EMT/inflammatory response.

### 3.6. Evaluation of Transcription Factor Binding Motifs in DARs Induced in Lens Epithelial Explants by the Addition of Vitreous Humor

Transcription factor binding within regions of open chromatin can drive specific changes in gene expression. With this in mind, we broadly divided the differentially accessible regions found between CM- and DM-treated explants into either promoter or non-promoter regions. We then searched these regions for transcription factor motifs using the monaLisa R package software.

We identified a total of 148 transcription factor motifs, using a significance cutoff of (*p* < 0.00005), within DARs of the DM-treated explants. Of these, a total of 24 and 57 motifs were enriched in the D1_DM and D5_DM non-promoter regions, respectively and 22 and 45 motifs enriched in the D1_DM and D5_DM promoter regions, respectively (Appendix A). A total of 18 of the 148 motifs were enriched in all the conditions (Appendix A). To prioritize the biological significance of these predicted transcription factor binding sites, we compared the list of 148 motifs with the transcription factors that exhibited differential expression in the DM-treated samples. This narrowed down the transcription factor transcripts to a list of 6 that were down-regulated and 11 that were up-regulated in DM (Figure 7A). Among the 6 down-regulated transcription factor genes, three were members of the Sox gene family (*Sox6*, *Sox17* and *Sox18*). The other three were *Fos, Aft3* and *Tead4.* Members of the Sox gene family were also prominent in the up-regulated transcription factors (*Sox3, Sox 4, Sox5, Sox9, Sox10* and *Sox11*), as were all three members of the Runx family (*Runx1*, *Runx2* and *Runx3*). The remaining two up-regulated transcription factor genes were *Tfap2c* and *Batf*. Of the transcription factor genes up-regulated in DM, only *Sox4* and *Sox5* exhibited high expression in the IMD (native lens epithelium).

We also examined our RNA-seq data set to identify transcription factor genes that were expressed, but not differentially regulated in any of our experimental conditions. Relying on a list of all transcription factor-encoding genes in the mouse genome [79], we identified 825 genes that met these criteria (Appendix A). Presumably, these genes represent a list of factors which are present in lens explants and may exert differential chromatin-binding activity but are not regulated at the transcript level. Interestingly, at the top of this list is *Pax6*, reflecting its high expression in lens cells. Moreover, near the top of the list are *Smad3*, *Smarcc1*, *Six3*, and *Tead2*. Motifs for these transcription factors that were over-represented in accessible chromatin in the D1_DM and D5_DM explants are annotated with an asterisk in Appendix A, respectively.

We independently repeated our explant experiments to validate our RNA-seq results on selected genes using RT-qPCR. As expected, we observed evidence for increased transcript abundance of fiber cell genes *Slc2a3* and *Caprin2* after 5 days of vitreous treatment, relative to the CM control. This experiment confirmed that *Runx1, Runx2, Runx3* and *Sox10* exhibited a significantly increased expression in the D5_DM condition relative to the native lens epithelium (IMD). All of these transcription factors, with the exception of *Runx2*, also exhibited a significantly increased expression in the D5_DM condition versus the D5_CM condition (Appendix A). Similarly, RT-qPCR provided evidence of a reduced *Tead4* abundance in D5_DM samples relative to D5_CM. Interestingly, RT-qPCR analysis of *Sox9* abundance failed to produce evidence of an elevated expression in response to vitreous treatment. Thus, the interpretation of our RNAseq results should be balanced with this observation, which could potentially be ascribed to differences between the assays, such as detection sensitivity, alternate cDNA generation approaches, or the reliance of RT-qPCR on a calibrator gene.

To compare the expression of these transcription factors in DM-treated explants with endogenous lens cells, we examined the expression of these 18 transcription factor genes from a recently published human snRNA-seq dataset from an adult anterior segment [51]. Nine of the eighteen transcription factor genes were expressed in at least 5% of the 13,900 lens cells in the human snRNA-seq dataset. In adult human lenses, *RUNX1* and *RUNX2* are more prominently expressed in fiber cells than in epithelial cells. In contrast, *TEAD4* and *FOS* are more prominently expressed in epithelial cells. *ATF3* expression appeared highest in cells of the transitional zone and early fiber differentiation. Of the Sox genes, only *SOX4, SOX5* and *SOX6* were prominently expressed in the adult lens data set (Appendix A). We examined the presence of the motifs for each of these nine transcription factor genes in the DARs from CM- and DM-treated samples (Figure 7B). We examined each motif in terms of increasing log_2_ fold change (LFC) value (left heat map) and adjusted *p*-value (right heat map). In each case, the D5_CM versus D5_DM DAR data are represented by different bins with the center bin of each heat map representing no LFC or the least significance, with LFC and significance increasing in each direction from the center for the left and right heat maps, respectively. Binding motifs for RUNX1, RUNX2, TEAD4, SOX6 and SOX4 were highly overrepresented amongst regions with increased accessibility in DM-treated samples at D5. In contrast, ATF3 motifs are more uniformly enriched in D5_DM DARs without respect to LFC.

We again utilized monaLisa to annotate the positions of the motifs for the nine transcription factors that were differentially expressed in the DM treatments, and significantly expressed in the adult human lens, in the DARs between the CM- and DM-treated samples. We then searched to determine if these annotated motifs were present in the promoter regions of genes characteristically expressed in lens fiber cells or lens epithelial cells (Figure 8, Appendix A). We found several TEAD motifs in gene promoter regions, including those for *Foxe3*, *Gja1* and *Dll1* (Figure 8A) genes that are primarily expressed in lens epithelial cells. We also found SOX and RUNX motifs in the promoters of several genes including *Tmod1*, *Dnase2b* and *Crybb3* (Figure 8B), and *Slc2a3*, *Nav3* and *Jag1* (Figure 8C), respectively; all genes expressed predominantly in lens fiber cells.

## 4. Discussion

The lens epithelial explant system has served as an in vitro model of lens fiber cell differentiation for nearly sixty years. While other methods of lens epithelial cell culture, notably dissociated primary cultures of embryonic chick lens epithelial cells, have proven useful for gaining insights into mechanisms of fiber cell differentiation [42,80,81], only the lens epithelial explant system maintains normal contact between the epithelial cells, as well as epithelial cell contact to their native basement membrane, the lens capsule. Studies in lens epithelial explants provided the fundamental framework for the theory that lens polarity relied on a gradient of FGF activity within the eye [28,82,83], a theory that later found support from studies in vivo [84,85,86,87]. While purified FGF reliably induces morphological changes and the expression of a set of proteins characteristic of fiber cell differentiation, other factors present in the ocular media (vitreous humor) undoubtedly support lens fiber cell differentiation [30,31,32,88]. Clearly, endogenous fiber cell differentiation follows stereotypical changes in chromatin organization that impact the lens transcriptome [48,49]. Here, we provide a comprehensive evaluation of how the addition of vitreous humor changes both the chromatin landscape and transcriptional profile in lens epithelial explants with respect to fiber cell differentiation.

While several genes associated with fiber cell differentiation were already accessible in lens epithelial cells, we observed a consistent trend of increased accessibility in the promoter regions of many such genes following the addition of vitreous humor. This increased accessibility was not observed in explants treated with unsupplemented culture media. Only rarely did we observe the addition of vitreous reducing promoter accessibility in genes associated with fiber differentiation. In contrast, the addition of vitreous had a less consistent effect on genes highly expressed in the lens epithelium. Often the accessibility of these genes remained the same or slightly decreased following the addition of vitreous. Overall, the majority of genes characteristic of both lens epithelial cells and lens fiber cells are accessible in the lens epithelium and either become more accessible (in the case of fiber genes) or change little (in the case of epithelial genes) upon the addition of vitreous. However, after one day of vitreous exposure (D1_DM), there was a characteristic drop in the level of transcripts for many fiber cell-associated genes (Figure 5B). This initial drop in differentiation-associated mRNA transcripts upon the addition of vitreous is reminiscent of the drop in crystallin transcripts seen in P3 rat lens epithelial explants 20 h after the addition of FGF2 [77]. This reduction was explained by increased mRNA turnover following FGF addition. A similar phenomenon may be responsible for the reduction in differentiation-related transcripts in lens explants 24 h after vitreous addition. At day 5, the increase in fiber differentiation-related transcripts was modest and often only slightly increased over that seen in the un-supplemented culture medium. This observation does not match the increased accessibility seen in many of these same genes in the D5_DM condition. We suggest that the explants treated with vitreous for only five days may not have had time to upregulate many of the differentiation-associated transcripts to the extent that we would see with longer culture. Previously, we found that vitreous exposure for 10 days led to a stronger expression of gamma crystallins and MIP protein than exposure for 5 days in P8 mouse lens epithelial explants [32]. It also is probable that the expression of many genes associated with fiber cell differentiation would follow a sustained increase in Erk signaling [30,41], and many of these genes increased from day 1 to day 5 (Appendix A). In contrast, the drop in the expression of genes associated with lens epithelium was dramatic at one day and maintained at five days of vitreous exposure (Figure 5C).

Based on many previous studies, we expected that the addition of vitreous humor would induce the expression of transcripts related to fiber cell differentiation. However, we did not anticipate the dramatic and widespread upregulation of immune system- and EMT-related transcripts upon the addition of vitreous. These upregulated transcripts were accompanied by significant increases in promoter accessibility in the presence of vitreous humor. In previous studies using lens epithelial explants to model fibrotic changes seen in posterior capsule opacification, TGFβ was often added to the culture medium as an inducing agent for EMT [89,90,91,92]. However, our observations suggest that the addition of vitreous is sufficient to induce many gene expression changes associated with EMT and fibrosis (Figure 5 and Appendix A). Interestingly, these genes are not typically expressed in the native lens epithelium and are not upregulated during endogenous fiber cell differentiation. This suggests that the in vitro culture conditions, combined with the addition of vitreous humor, ectopically upregulate the expression of these immune-/ECM-related genes. While the vitreous humor is in contact with the posterior lens fiber cells, the anterior lens epithelial cells do not normally come into direct contact with vitreous. It is possible that the combination of in vitro culture and the potential activation (via post-mortem surgical collection) of inflammatory mediators, including TGFβ, could amplify an immune response in the explants exposed to vitreous. Alternatively, the expression of these inflammation-related genes in response to vitreous may be suppressed within the intact eye.

Others have used the lens epithelial explant system to model pathogenic changes which occur in cataract [33,34] and posterior capsular opacification (PCO) [35,36,37,38,39,40,41,42,43]. Since the creation of lens epithelial explants involves traumatic removal of the fiber cell mass and in vitro culture, it is likely that explanting alone causes gene expression changes consistent with inflammation or trauma. A transcriptomic analysis of mouse lens epithelial cells in a cataract surgery model demonstrated many gene expression changes consistent with inflammation and fibrosis 24 h after the removal of the fiber cell mass [44]. However, our results demonstrate that the addition of vitreous humor enhances the expression of EMT/inflammatory markers over that of just explanting alone. In fact, the addition of vitreous humor to explanted lens epithelium may mimic the conditions that surviving lens epithelial cells encounter following cataract surgery, that leads to pathologic PCO.

To our knowledge, this work is the first to describe both genome-wide changes in gene expression and chromatin accessibility in lens epithelial explants induced to differentiate in vitro. One of the advantages of this approach is to make predictions about the transcription factors that may drive the observed changes in gene expression. We undertook an analysis of DARs to search for transcription factor binding motifs that are present in promoter regions of genes associated with either fiber cell differentiation or lens epithelial cells. After filtering the motifs identified in the promoter regions for transcription factors that displayed differential expression in the DM-treated samples, we identified six downregulated and twelve upregulated transcription factors. We then searched for these eighteen motifs in genes of interest. This analysis identified TEAD (a downregulated transcription factor) motifs in several lens epithelial genes (*Foxe3*, *Gja1* and *Dll1*) that decrease their level of expression with DM exposure. Likewise, we discovered SOX motifs in promoters of genes, including *Tmod1*, *Dnase2b* and *Crybb3* and RUNX motifs in the promoters of genes including *Slc2a3*, *Nav3* and *Jag1*, all of which are upregulated in fiber cells. Although the transcription factors PAX6, FOXE3 and PROX1 clearly play unique and important roles in lens development, we found no evidence that the DM treatment induced accessibility changes in binding motifs for these sites relative to the CM treatment using our stringent *p*-value criterion (*p* < 0.00005). We should note that the JASPAR 2022 vertebrate database of transcription factor binding profiles that we used [75] does not contain a FOX3 binding motif, but none of the FOX-family binding motifs present in the database showed evidence for DM-induced accessibility changes.

SOX transcription factors clearly play an important role in lens development. Distinct Sox genes are required both for lens induction (e.g., *Sox2*) [93,94] and for fiber cell differentiation (e.g., *Sox1*) [95]. SOX1 directly interacts with a promoter element in the gamma crystallin gene cluster and in its absence fiber cells fail to elongate and gamma crystallin expression is abolished [96]. While neither *Sox1* nor *Sox2* appear to be upregulated within five days of vitreous exposure, other Sox transcription factor genes (*Sox9*, *Sox10* and *Sox11*) are, and may be recruited to cis-regulatory sites to drive fiber cell differentiation in the explants. SOX transcription factors can also interact with other transcription factors such as PAX6 [95] or POU2f1 [97] to regulate lens gene expression. The *Dnase2b* promoter may provide an example of this type of cis-regulatory grammar as the accessible regions that we identified contain HSF4 and PAX6 consensus motifs, both of which have been previously suggested to regulate transcription at this locus [98], as well as motifs for SOX transcription factors (Appendix A).

In contrast to SOX transcription factors, there exists little information to suggest that RUNX transcription factors play an essential role in lens development. Mice lacking *Runx1* die as embryos from hematopoietic failure [99,100] and mice lacking *Runx2* exhibit neonatal lethality from respiratory failure [101,102]. *Runx3* knockout mice have a variety of phenotypes involving the nervous, digestive, immune, skeletal and respiratory systems [103,104,105,106,107]. Although *Runx1* and *Runx3* play a role in lacrimal gland morphogenesis and regeneration [108], functions for these transcription factors in lens development remain unknown, despite the expression of *RUNX1* in maturing fiber cells of adult human lenses [51]. RUNX motifs were also previously identified as uniquely enriched in the open chromatin peaks from mouse E14.5 lens fibers relative to lens epithelium [48]. RUNX transcription factors are involved in hematopoiesis and immune responses. Given the extensive upregulation of immune-related genes upon the addition of vitreous, it is possible that increased expression of RUNX transcription factors may facilitate lens fiber cell differentiation in vitro in a way that would not normally occur in vivo. The specific role of RUNX transcription factors in lens development remains to be experimentally verified.

The motifs for ATF3, and AP1 transcription factor family members in general, are dramatically enriched in chromatin that becomes accessible upon the addition of vitreous humor (Figure 7, Appendix A). These basic region leucine zipper transcription factors exhibit considerable ability to form heterodimers that bind similar cAMP-responsive (CRE), TPA-responsive (TRE) or MAF-recognition (MARE) elements [109,110]. ATF4 is essential for lens fiber formation [111] and is upregulated in several different rodent cataract models where it plays a role in ER stress/unfolded protein response [112,113,114,115,116]. Although the ATF3 motif was specifically highlighted in Figure 7, our experimental approach does not delineate which specific ATF family members are driving the chromatin accessibility changes induced by vitreous exposure.

Large changes in chromatin accessibility, and subsequent responses in gene expression, are likely driven by nucleosome repositioning or other epigenetic changes driven by chromatin remodeling enzymes. Unsurprisingly, some of these chromatin remodeling enzymes exhibit differential expression in response to vitreous. For example, *Prdm13, Tet1, Hdac11, Prdm9,* and *Tet3* are differentially upregulated in DEG clusters 3 or 6, characteristic of the native lens epithelium. However, other epigenetic remodeling enzymes exhibited increased expression in response to vitreous. Some of these include *Hdac9, Dnmt1,* and *Dntm3l* which are upregulated in DEG clusters 1, 2, and 4, respectively.

Lens epithelial explants have provided a way to model lens fiber cell differentiation in vitro for decades [19]. Within this system both FGF [29,83,117,118,119,120,121,122,123] and vitreous humor [30,117] have been shown to induce proliferation, elongation and the expression of several proteins characteristic of lens fiber cells. These changes in lens epithelial cell behavior are thought to result from the activation of both ERK and AKT [124]. In fact, other signaling pathways that can promote ERK and AKT activation can induce fiber cell differentiation independently of FGFR signaling [32,125]. Here, for the first time, we have provided a genome-wide comprehensive evaluation of vitreous-induced lens fiber differentiation in lens epithelial explants that includes both changes in chromatin architecture and changes in gene expression. We conclude that vitreous induces increased accessibility in the promoters of many genes associated with fiber cell differentiation and with the activation of immune responses and epithelial to mesenchymal transition. Vitreous also induces the suppression of transcripts characteristic of lens epithelial cells and increases in transcripts associated with fiber cell differentiation and fibrotic responses. The increased promoter accessibility in immune- and ECM-related genes appears to be a feature of the lens epithelial explant system that is not shared by endogenous lens fiber cell differentiation.

## Figures and Tables

**Figure 1 cells-12-00501-f001:**
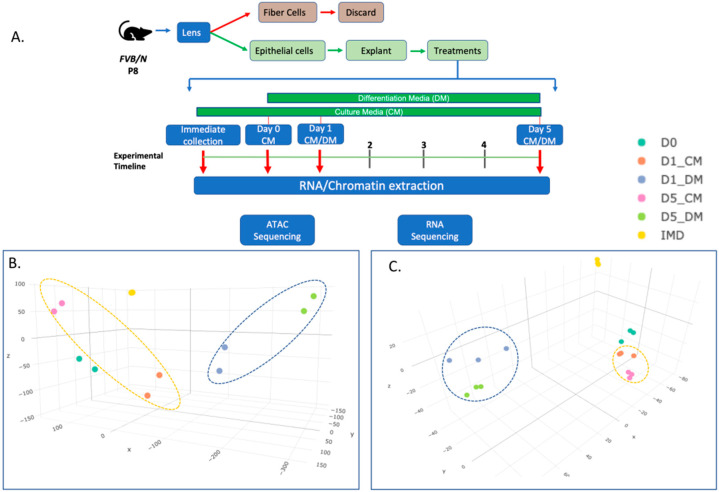
Profiling chromatin accessibility and gene expression in lens epithelial explants. (**A**) Overall experimental procedure. (**B**,**C**) 3-dimensional principal component analysis of ATAC- seq (**B**), and RNA-seq (**C**) summarizes the variation present in normalized chromatin accessibility and gene expression values across six conditions. The yellow dashed oval encompasses the variation for culture media as a function of time, whereas the blue dashed oval encompasses the variation for differentiation media as a function of time.

**Figure 2 cells-12-00501-f002:**
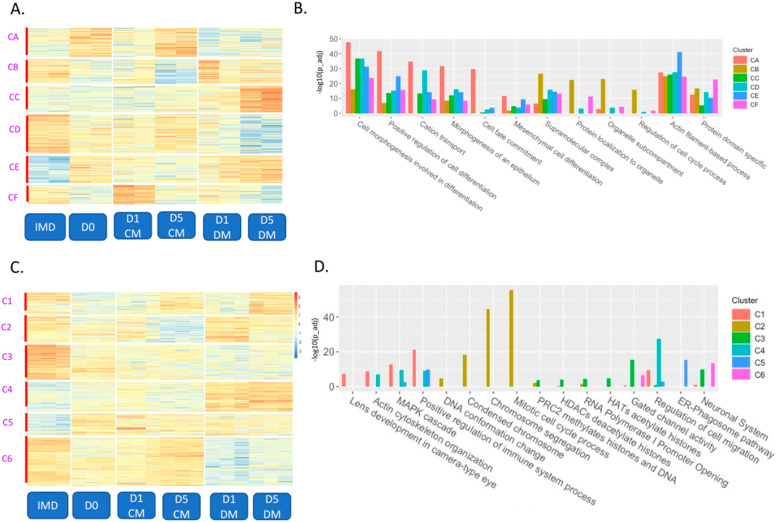
Clustering of differentially accessible regions and differentially expressed genes reveals a distinct signature of chromatin landscape and transcriptome as a function of time and culture conditions. (**A**) K-mean clustering was performed on all DARs across all fifteen pairwise comparisons (IMD vs D0, IMD_vs_D1_CM, IMD_vs_D5_CM, IMD vs D1_DM, IMD vs D5_DM, D0 vs D1_CM, D0 vs D5_CM, D0 vs D1_DM, D0 vs D5_DM, D1_CM vs D5_CM, D1_CM vs _D1_DM, D1_DM vs D5_DM, D1_CM_vs_D5_DM, D1_DM_vs_D5_CM, D5_DM vs D5_CM) by the criteria log_2_ fold change ≥ 1.5 and adjusted *p*-value (*p*-adjust) of ≤0.05 (*n* = 129,635 DARs), resulting in six clusters. (**B**) The bar chart displays select gene ontology terms associated with each cluster for ATAC-seq. The negative log_10_ of the *p*-value for each selected GO term is plotted on the y-axis, with each cluster represented by a different color. (**C**) K-mean clustering was performed on all DEGs across all fifteen pairwise comparisons by the criteria log_2_ fold change ≥1.5 and adjusted *p*-value (*p*-adjust) of ≤0.05 (*n* = 9044 DEGs), resulting in six clusters. (**D**) Bar chart displaying the significance (−log_10_
*p*-value was plotted) of DEGs for selected GO terms associated with each cluster represented by different colors.

**Figure 3 cells-12-00501-f003:**
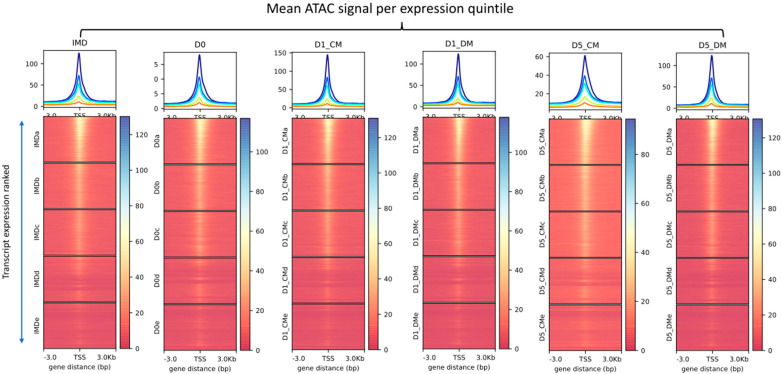
Relationship between transcriptional output and TSS chromatin accessibility across all conditions. The replicates for both RNAseq and ATACseq samples were collapsed for analysis. The highly expressed transcripts were ordered and divided into quintiles based on gene expression levels. The mean change in accessibility signal for each quintile is represented with different colored lines in the line graphs. Top panels: dark blue = 1st quintile, light blue = 2nd quintile, green = 3rd quintile, yellow = 4th quintile, orange = 5th quintile.

**Figure 4 cells-12-00501-f004:**
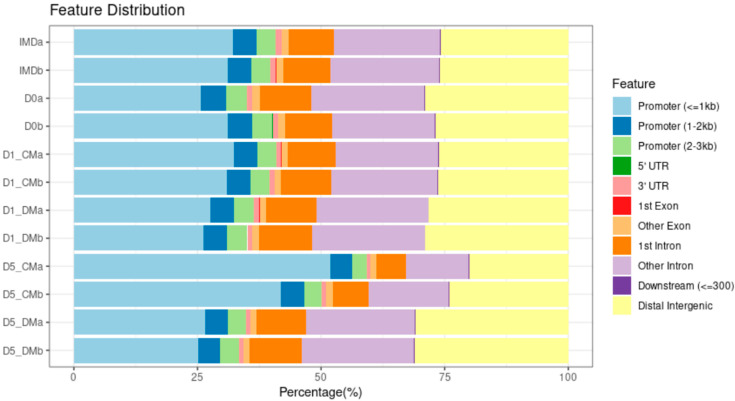
Gene annotation of all accessible regions for ATAC-seq using Chipseeker across six conditions. The total accessible regions are represented as a percentage to compare the proportion of changes in genomic features. The accessible regions are divided into eleven categories 1. proximal promoters (within 1 kb of the TSS), 2. intermediate promoters (within 1–2 kb of the TSS), 3. distal promoters (within 2–3 kb of the TSS), 4. first exon, 5. other exons, 6. first intron, 7. other introns, 8. 5’ untranslated region (5’ UTR), 9. 3’ untranslated region (3’ UTR), 10. downstream intergenic (within 300 kb of the 3’UTR), and 11. distal intergenic (>300 kb of the 3’UTR) regions.

**Figure 5 cells-12-00501-f005:**
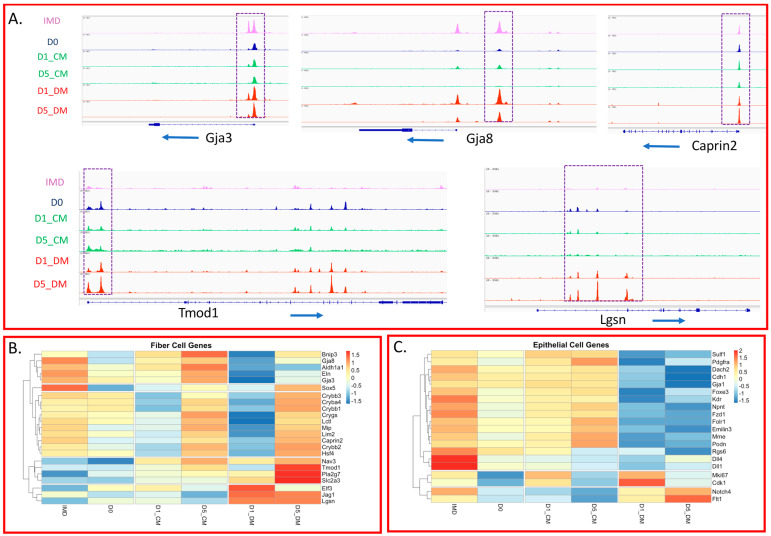
Fiber cell genes are active in D5 samples whereas epithelial cell genes are downregulated in DM samples. (**A**) Genomic browser view display of *Gja3*, *Gja8*, *Caprin2*, *Tmod1*, and *Lgsn* with normalized counts for ATAC-seq in the coverage tracks above. The dotted line represents the area of accessibility in promoter regions. (**B**) The z-score heatmap displays RNA-seq expression values for fiber cell-specific genes. (**C**) The z-score heatmap displays RNA-seq expression values for epithelial cell-specific genes.

**Figure 6 cells-12-00501-f006:**
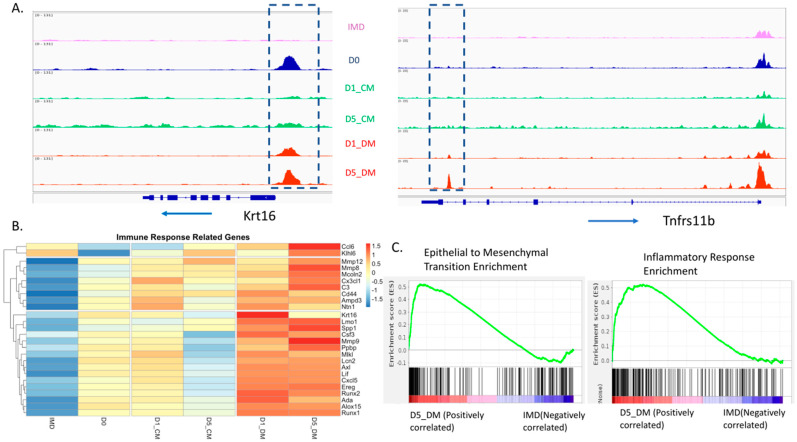
Immune response and epithelial to mesenchymal transition were observed in D5_DM samples compared to any other condition. (**A**) Genomic browser view display of *Krt16* and *Tnfrs11b* with normalized counts for ATAC-seq in the coverage tracks above. The dotted line represents the area of accessibility in promoter regions. (**B**) The z-score heatmap displays RNA-seq expression values for immune response-related genes. (**C**) Gene set enrichment analysis (GSEA) was performed on normalized RNA-seq expression values from IMD and D5_DM samples. The vertical black lines represent the gene in the query list, whereas the green line represents the enrichment score for each gene (vertical black line). D5_DM dataset (represented with a red scale with red being D5_DM and blue being IMD) is enriched for epithelial to mesenchymal transition and inflammatory response. The normalized enrichment score for Epithelial to Mesenchymal Transition and Inflammatory Response is 1.29 and 1.25 respectively.

**Figure 7 cells-12-00501-f007:**
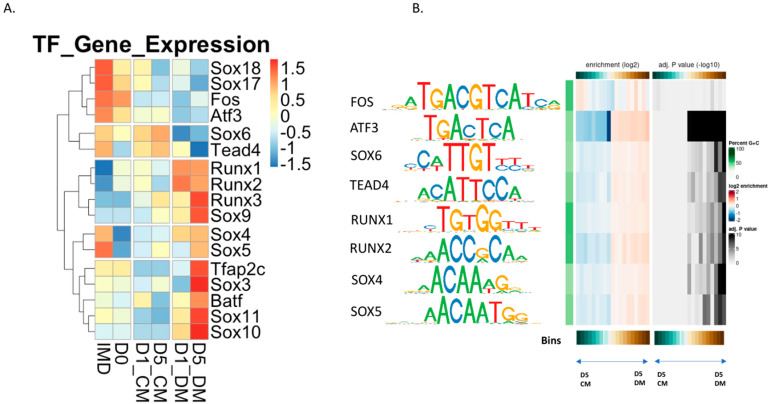
Overrepresented motifs in accessible regions (both promoter and non-promoter regions) at day 5. The monaLisa software was used to bin differentially accessible summits by accessibility across D5_DM and D5_CM conditions. Each bin was analyzed for the overrepresentation of the transcription factors motif using JASPAR 2022 database. Enriched motifs identified at −log10(0.00005) in bins were recorded. The associated transcription factor’s gene expression of the identified motifs is represented as a z-score heatmap (**A**). (**B**) The selected motifs enriched in D5_DM with increased transcript expression in endogenous human single-cell lens data (see Appendix A) are represented and plotted. The green heatmap indicates GC content. The blue and red heatmap represents the motif enrichment whereas white and black heatmap represents the adj. *p*-value. The center bin represents the zero enrichment and non-significant, whereas the left and right bins represent the progressively increased log fold change values, with the end value being the highest log-fold change.

**Figure 8 cells-12-00501-f008:**
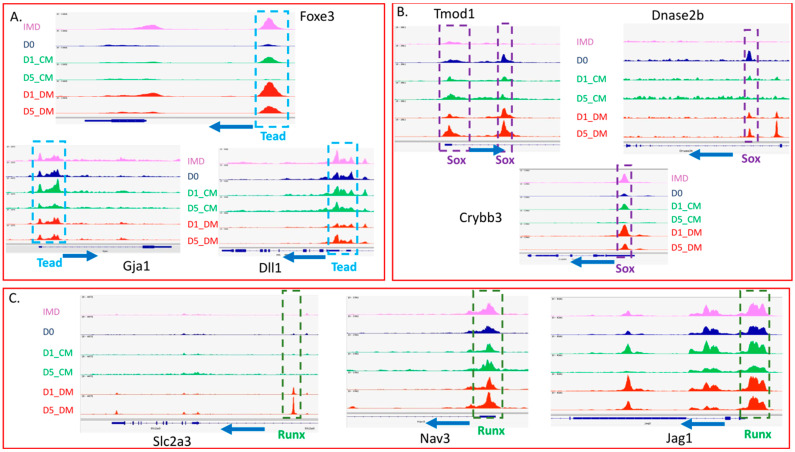
The genomic browser displays ATAC-signal proximal to DARs containing associated motifs. The dotted line represents the regions in the promoter of the representative genes with the motif sequence. (**A**) TEAD binding motifs for epithelial cell-specific genes *Foxe3*, *Gja1*, *Dll1*. (**B**) SOX binding motifs for fiber-specific genes *Tmod1*, *Dnase2b*, *Crybb3*. (**C**) RUNX binding motifs for fiber-specific genes *Slc2a3*, *Nav3*, *Jag1*.

**Table 1 cells-12-00501-t001:** The six clusters of ATAC-seq and six clusters of RNA-seq were compared to determine if specific clusters of DARs corresponded to specific DEG clusters. The highlighted boxes represent clusters that are associated with each other based on the permutation method described in Section 2.9. CA–CF represent ATAC-seq clusters and C1–C6 represent RNA seq clusters.

	CA	CB	CC	CD	CE	CF
**C1**	267	157	153	227	67	37
**C2**	300	163	129	198	107	66
**C3**	433	198	152	417	78	91
**C4**	395	141	236	166	269	46
**C5**	296	96	106	124	152	36
**C6**	487	434	217	514	119	119

**Table 2 cells-12-00501-t002:** All the accessible regions were annotated to the nearest promoter and analyzed for the common changes in the chromatin accessibility with changes in gene expression. ATAC-seq CM up represents the accessible regions for culture media treated samples with increased accessibility. Similarly, ATAC-seq DM up represents increased accessibility for differentiation media. RNA-seq CM up represents genes that are upregulated in culture media and RNA seq DM up genes that are upregulated in differentiation media.

	ATAC-Seq CM_D1 Up	ATAC-Seq No Difference	ATAC-Seq DM_D1_Up
RNA seq CM_1_up	12	3382	193
RNA seq no difference	61	9180	405
RNA seq DM_1_up	26	4221	145
	**ATAC-Seq CM_D5_Up**	**ATAC-Seq no Difference**	**ATAC-Seq DM_D5_Up**
RNA seq CM_D5_up	636	2434	915
RNA seq no difference	1848	5881	2580
RNA seq DM_D5_up	593	2544	1131

## Data Availability

The data discussed in this publication have been deposited in NCBIs Gene Expression Omnibus (Edgar et al., 2002: https://doi.org/10.1093/nar/30.1.207) and are accessible through GEO Series accession number GSE215953 (https://www.ncbi.nlm.nih.gov/geo/query/acc.cgi?acc=GSE215953). All code will be made available upon reasonable request to the authors.

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
