# Peer review of "Lens Epithelial Explants Treated with Vitreous Humor Undergo Alterations in Chromatin Landscape with Concurrent Activation of Genes Associated with Fiber Cell Differentiation and Innate Immune Response"

_cells, 2023, doi:10.3390/cells12030501_

Round 1

Reviewer 1 Report

The manuscript by Anil Upreti and co-workers provides novel insights into the molecular mechanisms of mammalian lens fiber cell differentiation. Despite of many earlier studies on this topic, implementation of multi –omics approaches allows generation of novel and unbiased insights into this critical process of lens morphogenesis and growth. This project is driven by using ATAC-seq in combination with CHIPseeker tool to determine dynamic changes in “open” and “closed” chromatin landscape in combination with bulk RNA-seq transcriptomic to identify differentially expressed genes (DEGs). As multiple treatments exist to induce lens fiber cell differentiation, the present manuscript employs treatment of postnatal day 8 (P8) lens epithelia from FVB/N mice with 50% bovine vitreous humor for one or five days as a model system. Biological replicates are included. Comprehensive data analyses include the primary data as well as earlier published databases. Many novel clusters of functionally related genes are reported here for the first time, such as transient downregulation of important regulatory and architectural lens proteins (Fig. 5) as well as upregulation of EMT and inflammatory response genes (Figs. 6 and S7). The main issue is that many figures are too small to appreciate without any magnifying glass (hard copy printed version) and should be enlarged. The manuscript will be 2-5 pages longer but much better readable without using the computer screen. Taken together, the present studies provide novel insights into the complex cellular and molecular processes governing lens fiber cell differentiation and are within the scope of this special issue of Cells

Additional comments:

1)   Results (p9, line 434): Edit sentence “Samples and was enriched for GO terms …”.

2)    Results/Motif analysis: It would be great to comment whether the package used contains any data on motifs recognized by Prox1 and FoxE3 transcription factors their absence might limit the analysis. 

3)    Likewise, any difference in ATF3 and ATF4 motifs as ATF4/CREB2 is important for lens development? 

4)    Page 17/line 824: typo Sox4.

5)    Given the importance of Dnase2b gene for the denucleation process and its highly lens-specific expression, Fig. 8 can be easily expanded to shown the novel Sox sites identified here together with all previously known binding sites, e.g. Hsf4, Pax6 and TBP/TATA-box. 

Reviewer 2 Report

This manuscript by Upreti et al. utilized a mouse lens epithelial cell explant model to identify effects of exogenously added bovine vitreous humor on cellular chromatin configurations and potentially related changes in gene expression. Lens epithelial explants prepared from P8 were cultured in media or media diluted 50% with bovine vitreous humor (“differentiation media”) for 1 and 5 days and ATACseq and RNAseq was employed to identify potential changes in chromatin accessibility and gene expression. Using k-means clustering, the authors found 6 cluster patterns of differentially altered chromatin and 6 cluster patterns of differentially expressed genes.   Some of these differentially accessible regions were highly correlated with induced expression of genes previously established to be expressed in lens fiber cells and while others were associated with some reduced expression of epithelial cell genes.  Collectively, these results implicate vitreous exposure with changes in chromatin that could play a role in regulating genes associated with lens fiber cell differentiation. Genes with expression patterns associated with 5 days exposure of lens explants to vitreous humor were analyzed by GO-type software and some were associated with epithelial-to-mesenchymal transition and inflammatory response. Sequence consensus analysis of the differentially accessible regions revealed the presence of transcription factor consensus sequences for SOX and RUNX family of transcription factors. The manuscript is well-written, the data is for the most part clear and rigorously analyzed.  The overall conclusion that vitreous exposure is associated with chromatin changes and altered gene expression in the model system is sound. There are a few minor questions and comments the authors should consider:

1.     Vitreous contains many growth factors previously shown to influence lens cell differentiation.  The authors should speculate on what these factors might be.

2.     Lens epithelial cells are not exposed normally to vitreous. How does this artificial exposure of vitreous mimic anything about the natural lens and its differentiation?  Does this model system really only apply to understanding PCO and if so it would help the reader to more clearly define and explain this.  

3.     Would adding aqueous instead of vitreous affect the parameters examined-could the authors speculate?

4.     Doesn’t FGF/IGF etc induce similar gene expression changes and shouldn’t these factors be examined for their ability to elicit “vitreous-specific” changes?

5.     The authors should include what software/programs were used for the k-means clustering and cluster permutation analysis.

6.     Since k-means clustering requires you to choose the number of k-clusters before initiating the clustering algorithm, how did the authors determine that 6 clusters was the optimum number of clusters for differentially accessible regions and differentially expressed genes?

7.     As stated by the authors, other studies have conducted RNAseq and ATACseq on lens differentiation models from mice and chicken (references 33, 36, and 37). Are there any similar results between the present study and these different model systems? Specifically, are there similar GO terms, fiber cell or epithelial cell genes, and predicted transcription factor consensus sequences found in the present study compared to these past studies? Common pathways and genes could further support the premise that the vitreous is important for regulation of gene expression associated with lens fiber cell differentiation.

8.     Some of the RUNX and or SOX family transcription factor expression levels should also be validated at the protein level via western blot.

9.     Figure 3 embedded in the manuscript does not appear to show the ATACseq signal profiles in the six plots arranged above the heatmaps. They appear to be blank in the manuscript.

10.  Based on the title and abstract, the authors imply that the immune response genes are of great importance in the results of this study. However, the authors do not seem to expand on these findings in the discussion very much. 

11.  Since many genes have multiple functions and multiple associated GO terms, do the specific genes in the present data that are associated with the GO term “immune response” have other general functions not related to the immune response? Could those non-immune response functions have a significant association with vitreous-dependent fiber cell differentiation?

12.  The authors show that some of the gene clusters contain genes associated with fiber cell gene expression from reference 33. Is there a statistically significant association between the vitreous dependent gene expression changes and fiber cell gene expression changes in reference 33? This could be determined by taking the list of genes that appear to be induced by vitreous treatment and performing a hypergeometric distribution test (or Chi-square/Fisher exact test) versus the genes classified as “fiber genes” from reference 33. This is similar to how various GO analysis programs determine significant GO terms. Additionally, the authors could take the list of genes repressed by vitreous treatment and statistically compare them to the list of genes classified as “epithelial genes” from reference 33.

13.  The authors provide a global pearson correlation analysis of gene expression changes versus chromatin accessibility changes (r = 0.404). However, since the authors have 6 treatment groups, they could also perform pearson correlation analysis on each individual gene to assess the relationship between chromatin accessibility changes and gene expression changes for each gene. You can limit this analysis to include only genes that are differentially expressed in at least one of the pairwise comparisons. For example, are there any individual genes differentially expressed in D5_DM vs IMD or differentially expressed in D1_DM vs IMD, that have a high correlation between chromatin accessibility and gene expression changes across all 6 treatment groups (r > 0.7)? This would reveal the genes with the greatest association between chromatin accessibility and gene expression changes and could reveal. The authors could also perform GO analysis on the genes with a high correlation between gene expression and chromatin accessibility. Something similar might have been attempted in the results described in Tables 1 and 2, but they appear to describe global associations rather patterns associated with specific genes.

14.  In the transcription factor binding motif analysis, the authors use the RNAseq data to identify differentially expressed transcription factors to narrow down the candidate transcription factors. However, it might also be interesting to separately include the highly expressed transcription factors that are not differentially expressed (i.e ones that have high expression levels and have relatively stable expression independent of vitreous). The binding activities of these vitreous-independent transcription factors would solely be regulated by chromatin accessibility changes and could provide an interesting finding.

15.  Do any of the differentially expressed genes have chromatin modifying activities? If so, the authors should discuss the potential role of these chromatin modifying enzymes on nucleosome repositioning or epigenetic changes.

Author Response

Please see attachement.
